# The Effects of Tourism Development on Eco-Environment Resilience and Its Spatio-Temporal Heterogeneity in the Yangtze River Economic Belt, China

**Kun Wang [1,2,\*]**, **Xiangtai Chen [1]**, **Zhenxian Lei [1]**, **Songxin Zhao [1]** and **Xiao Zhou [1,2]**

1   College of Tourism and Culture Industry, Guizhou University, Guiyang 550025, China; gs.xtchen21@gzu.edu.cn (X.C.); gs.zxlei20@gzu.edu.cn (Z.L.); gs.sxzhao22@gzu.edu.cn (S.Z.); xiaozhou@gzu.edu.cn (X.Z.)
2   Culture and Tourism Economic Quality Development Operation Center, Guizhou University, Guiyang 550025, China
\*   Correspondence: kwang3@gzu.edu.cn

**Abstract:** Tourism sustainability is a significant approach to forming a synergistic model of industry and ecology in ecologically vulnerable areas. Scientifically detecting the effect mechanism of tourism development on eco-environment resilience is important in achieving regional social-ecological system sustainability. In this work, empirical exploration is conducted on the tourism development index (TDI) and eco-environment resilience index (ERI) in the Yangtze River Economic Belt (YREB) to study the spatio-temporal heterogeneity of TDI's effect on the ERI. The results indicate significant growth in the TDI in the YREB, with the formation of tourist clusters around Shanghai and Chongqing as the core. Although the ERI typically exhibits a declining trend, the rate of decline has notably slowed, forming a "high at the sides and low in the middle" spatial pattern. The TDI and ERI are spatially dependent in the YREB, with predominantly high-high (HH) and low-high (LH) clusters in Shanghai, Zhejiang, and Jiangsu. Conversely, upstream regions with strong eco-environmental foundations exhibit low-low (LL) and high-low (HL) clusters. In general, the TDI promotes the ERI, but there is significant spatio-temporal heterogeneity in the YREB. Positive impact regions are expanding, while negative impact regions are shrinking. These results could provide scientific evidence for differentiated classification and control policies in the YREB.

**Keywords:** tourism development; eco-environment resilience; spatio-temporal heterogeneity; Yangtze River Economic Belt of China

## 1. Introduction

The eco-environment system is an artificial ecological system gradually formed by urban and rural residents in the process of adaptation, production, and creation to the natural environment [1]. Under the impact of multiple external environments, such as globalization, urbanization, industrialization, and natural disasters, the urban eco-environment system is facing a series of issues, such as increasing environmental risks, frequent resource shortages, and ecological degradation [2–5]. Therefore, how to enhance the eco-environment resilience (ERI) in the face of external shocks has become an important element of sustainable urban development [6].

Tourism is an important support industry and a highly relevant industry in tourist destination cities, and it is a new driving force for urban economic transformation and high-quality development [7]. Urban tourism development and the ecological environment system are interrelated systems with regional and comprehensive characteristics under the specific human–land relationship, within which there is an obvious dualistic and contradictory relationship [8]. While the growth of the tourism economy brings economic benefits to cities, it may also be at the expense of the ecological environment. Therefore,

an in-depth investigation of the spatial relationship and its influence mechanism between the level of urban tourism development (TDI) and the ERI is not only conducive to the macro-control of the direction of rational regulation of regional tourism development, but also of strategic significance for the realization of a harmonious co-existence of tourism and the ecological environment.

This paper aims to construct a comprehensive evaluation index system of the urban TDI and ERI, and—with the help of geographic research methods such as bivariate spatial autocorrelation, the spatial econometric model, and the geographically weighted regression model—to explore the comprehensive impact of the TDI on the ERI and reveal the spatio-temporal heterogeneity characteristics of the impact, with a view to providing countermeasure suggestions for the co-ordinated development of the regional tourism industry and the ecological environment system.

The remainder of this paper is organized as follows: part two is a literature review; part three includes the construction of the TDI and ERI evaluation index system, methodological overview, and data sources; part four analyzes the calculation results, including the spatial distribution characteristics, bivariate spatial correlation characteristics, the comprehensive impact analysis, and its spatio-temporal heterogeneity analysis; part five puts forward the main conclusions and recommendations.

## 2. Literature Review

### 2.1. Impact of Tourism Development on Eco-Environment System

The relationship between tourism and the eco-environment system has been a hotspot explored by a number of scholars [9–14], and its research mainly focuses on the impact of tourism industry development on the ecological environment system; however, the degree of impact and whether the direction of the impact is positive or negative is not yet conclusive.

#### 2.1.1. Negative Impact

The development of tourism requires the support of the transportation industry, and the construction and operation of tourism transportation infrastructure, such as airports, highways, and cruise ship, depends on the use of energy resources; additionally, the expansion of the transportation industry is a major cause of environmental pollution [15,16]. The sustainable planning and management of tourism destinations is an important measure to ensure the benign development of the ecosystem of the destination [9], and illogical planning, development, and management of tourism destinations can cause the deterioration of the tourism ecosystem found at the destination [17,18].

The influx of tourists also exerts pressure on the ecological system of tourist destinations [19], and scholars have primarily focused on studying the impacts of trampling caused by tourists. The relevant research primarily centers around the repercussions of various types of trampling on both soil and plants. Scholars have employed comparative analysis through field surveys in the tourism domain or conducted numerous trampling experiments to accurately investigate the effects of trampling. This includes examining the influence of different types of trampling, such as pedestrians, horses, camels, various vehicles, etc., as well as the impact of different trampling intensities on plants and soil. Furthermore, researchers have explored the responses of different plant species and environments to trampling [10,19–21].

In addition, the influx of tourists also brings about energy consumption and increased carbon dioxide emissions in tourist destinations, and there is evidence that tourism activities are detrimental to the quality of the air environment. In addition, the expansion of the tourist scale is also an important influence on climate change [12,22,23]. The concept of over-tourism has also been mentioned by scholars, where a large influx of tourists can lead to overcrowding in tourist destinations, exceeding the ecological capacity of the destination and causing damage to its ecosystem [24,25].

### 2.1.2. Positive Impact

The development of tourism also positively affects the ecosystems of tourist destinations. The arrival of tourists facilitates the flow of capital and increases tourism revenue in tourist destinations, which in turn promotes the economic growth of tourist destinations [26]. This not only improves the infrastructure of tourist destinations and provides a large number of employment opportunities, but also provides financial and policy support for the improvement of the ecological environment of tourist destinations, which is conducive to the optimization of the ecological system of tourist destinations [27].

In addition, the development of tourism will also promote the optimization of the industrial structure of tourist destinations [28] and promote the transformation of tourist destinations from reducing high-energy-consuming and high-polluting industries to high-energy-saving and low-polluting industries [29], which is conducive to the improvement of the ecological environment system of tourist destinations [30,31]. The benign development of tourism will also lead to increased investment in tourism projects in tourist destinations. Alam et al. (2017) empirically explored the correlation between tourism investment and $CO_2$ emissions and concluded that tourism investment improves the quality of ecosystems by decreasing $CO_2$ emissions in tourism destinations [32].

### 2.2. Urban Ecosystem Resilience

#### 2.2.1. Resilience

Resilience is a crucial concept that has emerged in various fields of study, including engineering mechanics, psychology, and ecology, among other disciplines. The term resilience comes from the Latin word "Resilio", which means a system's ability to maintain stability and carry out core functions even after being subjected to external shocks [33]. Initially, resilience was used to address a material's ability to return to its initial state after being subjected to external forces [34], but it has since been extended to address complex systems such as socio-economic systems [35]. The concept of resilience has evolved from engineering resilience to ecological resilience, and finally to evolutionary resilience [33]. In general, resilience highlights a system's ability to resist, adapt, and recover from internal and external shocks [36].

#### 2.2.2. Urban Resilience

Urban resilience is a concept that has gained significant attention and research focus since its introduction by the International Council for Sustainable Regional Development (ICLEI) in 2002 [33]. It has been studied extensively in various disciplines, such as geography, ecology, and social sciences. The theory and practice of urban resilience aim to provide holistic and process-regulated solutions to address urban problems and promote sustainable urban development [37].

Different scholars approach urban resilience research based on their disciplinary backgrounds [38]. Geographers examine the resilience of urban systems from the perspective of natural disasters, while social scientists focus on the adaptive and coping capacity of urban communities in response to social, political, and environmental changes. Environmental scientists explore the extent to which urban systems can withstand external disturbances, and business management professionals explore the resilience of cities from the perspective of urban management [38–40]. Although a unified definition of urban resilience has not yet emerged [40,41], there is general agreement that it encompasses the concepts of the resistance, adaptation, organizational learning, and recovery of urban systems in response to various disturbances and stresses. The ultimate goal of urban resilience is to contribute to the sustainable development of cities by enabling them to maintain their original structure after being disturbed [42,43].

#### 2.2.3. Urban Ecosystem Resilience

The concept of urban ecosystem resilience is dependent on the connotation of resilience and urban resilience [43]. Specifically, in the development of resilience theory to the

third stage—evolutionary resilience—resilience focuses on the ability of things to absorb disturbances, rather than the pursuit of equilibrium [44]. Urban resilience is the evolution of the idea of resilience for urban system construction: with the city as a composite system with economic, social, and ecological subsystems, the concept of urban resilience can naturally be inherited into urban ecosystem resilience; that is, the resilience of the urban ecosystem system to deal with internal and external disturbances in response to the presentation of the coping, adaptive, and restorative capacity [45,46].

Urban ecosystem resilience has received wide attention from academics as an important dimension of urban resilience [47]. The relevant research has mainly focused on the assessment and influencing factors of urban ecosystem resilience [48,49]. This comprehensive assessment is an important part of the scientific cognition of urban ecosystem resilience. There is no unified assessment standard and research paradigm [38,50], and most of the studies start from the basic characteristics of urban ecosystem resilience and build a comprehensive assessment framework of urban ecosystem resilience in terms of the resistance, adaptability, and resilience of urban ecosystems in response to disturbances or risks [45–47]. The influencing factors of urban ecosystem resilience are diverse, and the mechanism of action is complex. The urbanization process has promoted the concentration of the population in cities, and the increase in urban population density has had a more obvious negative effect on the carrying capacity of its eco-environment [2,51]. Industrial transformation, technological innovation, and environmental policies will continuously promote the quality of urban ecosystems and urban ecosystem resilience [3,47,48].

## 3. Research Design

### 3.1. Study Area

The YREB, spanning eastern, central, and western China, can be divided into three sections: downstream, midstream, and upstream. It includes 11 provinces, such as Shanghai, Zhejiang, Hubei, Chongqing, and Yunnan, with a land area of about 2.05 million km² [11], and is a pioneering demonstration belt for China's economy and ecological civilization (Figure 1). Among them, the downstream of the YREB includes four provinces and cities in Shanghai, Jiangsu, Zhejiang, and Anhui; the midstream includes three provinces and cities in Jiangxi, Hubei and Hunan; and the upstream area includes four provinces and cities in Chongqing, Sichuan, Guizhou, and Yunnan. The implementation of the "Outline of YREB Development Plan" by the State Council marked that the YREB has been officially elevated to a national strategy. Therefore, the selection of the YREB as a case site is highly typical.

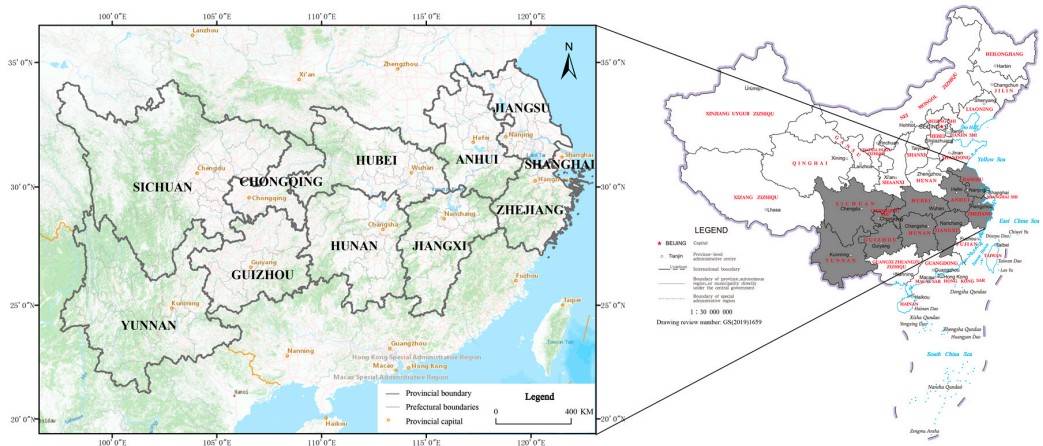

**Figure 1.** A general overview of the YREB.

With its dense population, highly developed urban agglomerations, and rich and diverse tourism resources, the YREB region is a major tourist destination and source of visitors in China. In 2019, the total number of tourists in the YREB reached 8.118 billion, accounting for 48.07% of the national proportion; the total tourism revenue achieved

was CNY 10.63 trillion, accounting for 47.70% of the national proportion. Meanwhile, there are 126 5A-level scenic spots in the YREB, accounting for 43.30% of the national proportion [52,53]. However, while the tourism economy is increasing rapidly, the hidden dangers of the ecosystem have not been eliminated. The rapid urbanization process alongside the high population concentration has put great stress on the regional eco-environment. Therefore, the YREB is representative as a case site for the study of the impact of the TDI on the ERI.

### 3.2. Data Sources

This paper takes 126 cities in the YREB as the objects. To ensure geo-spatial integrity, Tianmen, Xiantao, Qianjiang, and Shen Longjia Forest Area administered by Hubei Province are also included in the study, totaling 130 administrative units. The required data are taken from the Statistical Yearbook or bulletins such as "China Regional Economic Statistical Yearbook", "China City Statistical Yearbook", and "China Environmental Statistical Yearbook".

The YREB has always taken the pursuit of economy speed as its primary goal, leading to prominent issues such as resource constraints and pollution intensification. In particular, the cyanobacterial pollution outbreak in Taihu Lake and Chaohu Lake in 2007 brought widespread attention to the protection of water resources in the YREB [54]; China formally proposed the construction goals of the YREB as a "green ecological corridor" in 2014. At the end of 2018, China further established the direction of promoting regional socio-economic development with "green and ecological" as the core [55]. Therefore, the years of 2007, 2013, and 2019 are selected as the timepoints in this paper.

### 3.3. Methods

#### 3.3.1. Evaluation Indicator System of TDI and ERI

The tourism sector, as a proto-typical modern service industry, plays a vital role in promoting the sustainability of cities. A comprehensive and sustainable evaluation of the TDI is critical to achieving this objective [56–59]. The sustainability of the TDI involves the gradual optimization and alignment of tourism supply and demand, as well as the interactive integration of tourists with the society and environment of the tourism destination that reflects Human–Land relations [60–62]. The TDI is mainly measured in three dimensions: tourism market scale, resources and products of tourism, and contribution of tourism. Among them, the dimension of the tourism market scale reflects the overall situation of the urban tourism market in terms of tourism revenue, scale of tourists, and per capita tourism consumption capacity; the dimension of resources and products of tourism reflects the overall situation of urban tourism resources in terms of the abundance of urban tourism resources, the scale of high-grade tourism resources, and the integration of cultural and tourism products; and the dimension of contribution of tourism reflects the contribution of urban tourism to the economy, employment, optimization of industrial structure, and growth of residents' income. The contribution of tourism dimension reflects the contribution of tourism to the economy, employment, industrial structure optimization, and residents' income growth.

The ERI is mainly evaluated according to three aspects: pressure and resistance, adjustment and adaptation capacity, and vulnerability and recovery capacity [22–25]. Among them, pressure and resistance reflect the active resistance ability of an urban ecological environment system to external pressure, mainly from urban population scale growth, land use intensity, economic growth intensity, wastewater and exhaust gas and other pollutant emissions, as well as other aspects of the selection of indicators. Adjustment and adaptability reflect the ability of the urban ecological environment system to adjust and actively adapt in the face of external risks or disturbances, mainly from the aspects of urban waste treatment, sewage treatment and solid waste utilization. Resilience and recovery express the resilience and recovery ability of the urban ecological environment system to restore itself to its original state when it suffers from external disturbances, and is mainly from

the aspects of urban air quality, environmental greenery, green recreational space, water resources carrying capacity, and environmental governance capacity.

Thus, adhering to the demands of scientific rigor, systematicity, and accessibility and considering prior research [22,25,30,47,63–66], this paper cherry-picked 26 indicators, including total tourism income, tourist attractions, land use intensity, greenery coverage, and population density, to construct an assessment system for the TDI and ERI. Refer to Table 1 for the complete assessment system.

**Table 1.** The assessment system of TDI and ERI.

| Target | Guideline (Weight) | Indicator | Indicator Description (Attribute) | Weight | Reference |
|---|---|---|---|---|---|
| Tourism development (TDI) | Tourism Market scale (0.3093) | X1 Total tourism income | Reflecting the economic condition of tourism (+) | 0.1511 | [64] |
| | | X2 Total tourist trips | Reflecting the scale of visitors (+) | 0.1139 | [64] |
| | | X3 Per capita tourist consumption | Per capita tourist consumption capacity (+) | 0.0443 | [64] |
| | Resources and products of tourism (0.3971) | X4 High-level tourist attraction | Expressed by the number of Grade 3A or above (+) | 0.1184 | [30] |
| | | X5 state-level tourism resources | The sum of National Forest Park, National Geopark, National Scenic Spot, and World Heritage Site (+) | 0.0759 | [30] |
| | | X6 National intangible cultural heritage | Represents the integration of urban culture and tourism resources (+) | 0.1181 | [30] |
| | | X7 Number of museums for 10,000 people | | 0.0846 | [30] |
| | Contribution of tourism (0.2936) | X8 Tourism Industry Dependency | Total tourism income/ GDP (+) | 0.0947 | [64] |
| | | X9 Elasticity of urban residents' tourism income | Reflects the contributions that tourism makes to the revenues of urban and rural residents (+) | 0.0796 | [64] |
| | | X10 Elasticity of rural residents' tourism income | | 0.0210 | [64] |
| | | X11 Ratio of employees of tertiary industry | Tourism's contribution to employment (+) | 0.0237 | [64] |
| | | X12 The proportion of tourism income in tertiary sector income | Tourism's contribution to the optimization of industrial structure (+) | 0.0746 | [64] |
| Resilience of eco-environment (ERI) | Pressure and resistance (0.5014) | Y1 Population density | The pressure of population size on the ecosystem (−) | 0.0409 | [47] |
| | | Y2 Economy density | Ecosystem perturbation by economic growth (−) | 0.1514 | [47] |
| | | Y3 Land use intensity | Area of built-up /Urban land area (−) | 0.0811 | [47] |
| | | Y4 Wastewater discharge intensity | The pressure of wastewater on the ecosystem (−) | 0.1040 | [47] |
| | | Y5 Exhaust emission intensity | Exhaust pressure on ecosystems (−) | 0.1240 | [47] |
| | Adjustment and adaptability (0.1945) | Y6 Harmless disposal rate of domestic waste | Adaptation of cities to ecosystem pressures through solid waste, domestic wastewater treatment, and waste utilization (+) | 0.0029 | [65] |
| | | Y7 Per capita domestic waste removal volume | | 0.1778 | [65] |
| | | Y8 The rate of domestic wastewater treatment | | 0.0064 | [65] |
| | | Y9 Usage rate of solid waste | | 0.0074 | [65] |
| | Flexibility and recovery (0.3041) | Y10 Excellent air quality rate | Expressed by the number of days to reach level 2 (+) | 0.0076 | [47] |
| | | Y11 The rate of greenery coverage in the built-up region | Indicates the greening of the city's environment (+) | 0.0042 | [47] |
| | | Y12 Park area per capita | Indicates the green leisure space of the city (+) | 0.0136 | [47] |
| | | Y13 Water resources per capita | Indicates the water carrying capacity (+) | 0.1925 | [47] |
| | | Y14 Investment of the Environment Fund as a percentage of financial expenditure | Indicates the environmental management level (+) | 0.0862 | [47,66] |

### 3.3.2. Comprehensive Assessment Model (CAM)

In this paper, multi-objective linear weighting is applied to build a CAM of the TDI and ERI. The method consists of three steps:

Step 1: As the units of each indicator data are inconsistent, the data require being normalized initially [63]. If the indicator is positive,

$$x'_{ij} = (x_{ij} - x_{jmin})/(x_{jmax} - x_{jmin}) \tag{1}$$

If the indicator is negative,

$$x'_{ij} = (x_{jmax} - x_{ij})/(x_{jmax} - x_{jmin}) \tag{2}$$

where years and indexes are represented by *i* and *j*, respectively.

Step 2: Reasonable determination of index weights is the basic premise of the assessment. The weights are established using the entropy method [65]:

$$p_{ij} = x'_{ij}/\sum_{j=1}^{n} x'_{ij} \tag{3}$$

$$e_j = -k\sum_{i=1}^{n} p_{ij} ln p_{ij}, k = 1/lnn \tag{4}$$

$$w_j = d_j/\sum_{j=1}^{m} d_j, \ d_j = 1 - e_j \tag{5}$$

where $e_j$ is the entropy of the index *j*; $w_j$ is the weight of index *j*.

Step 3: Calculate the assessment value of the TDI and ERI through multi-objective linear weighting. The expressed formula is [67–69]:

$$Y = \sum_{i=1}^{n} w_j x'_{ij} \times 100 \tag{6}$$

where *Y* is the assessed value of the TDI and ERI.

### 3.3.3. Bivariate Spatial Autocorrelation Analysis (BISA)

The BISA could usefully detect the spatial correlation characteristics of multiple geographic variables, which is divided into bivariate global and local spatial autocorrelation (BI-GMSA and BI-LISA) [66,70]. This paper uses BISA to reveal the spatial dependence of the TDI and ERI from the global and local perspectives.

$$I_G = \sum_{i=1}^{n} \sum_{j \neq i}^{n} w_j \left( X_i^k - \overline{X^k} \right) \left( X_j^l - \overline{X^l} \right) / S^2 \sum_{i=1}^{n} \sum_{j \neq i}^{n} w_{ij} \tag{7}$$

$$I_L = \frac{X_i^k - \overline{X^k}}{\sigma^k} \sum_{j \neq i}^{n} w_{ij} \left( \frac{X_j^l - \overline{X^l}}{\sigma^l} \right) \tag{8}$$

where $I_G$ and $I_L$ are the Global and Local Moran Index of the TDI and ERI, respectively; $X_i^k$ and $X_j^l$ represent the values of the TDI and ERI, respectively; $\sigma^k$ and $\sigma^l$ are the exponential variances and *w* is the weight.

### 3.3.4. Spatial Econometric Model

The Spatial Econometric Models are the incorporation of spatial factors into an econometric regression model that captures the spatial interactions of geographic phenomena [71], including the SLM, SEM, SEMLD model, etc. The SLM model includes the spatial correlation of dependent variables; the SEM model incorporates the spatial relation into the error term and emphasizes the influence of error shock; the SEMLD model considers both

the spatial relationship of the explained variable and the extrinsic association of the error term [72,73].

The equation of SLM model is:

$$Y = \rho W y + X\beta + \varepsilon \tag{9}$$

The equation of SEM model is:

$$Y = X\beta + \varepsilon; \varepsilon = \lambda W \varepsilon + \mu \tag{10}$$

The equation of SEMLD model is:

$$Y = \rho W y + X\beta + \varepsilon; \varepsilon = \lambda W \varepsilon + \mu \tag{11}$$

where $X$ and $Y$ are the independent and dependent variables, respectively; $\rho$ and $\lambda$ are the coefficients of the spatial lag and error, respectively. $B$ is the estimation coefficient; $\varepsilon$ is the error vector.

### 3.3.5. Geographically Weighted Regression (GWR)

GWR can spatially estimate parameters for each geographic space, which can carry out local regression estimation based on geographical location changes and can better reflect the spatial correlation and dependence of geographical units [74–76]. The equation is as follows:

$$y_i = \beta_0(\mu_i, v_i) + \sum_{k=1}^{n} \beta_k(\mu_i, v_i) x_{ik} + \varepsilon_i \tag{12}$$

where $(u_i, v_i)$ is the lat/long co-ordinates; $\beta_k(u_i, v_i)$ is the coeff. Of regression.

## 4. Results

### *4.1. Spatio-Temporal Characteristics of the TDI and ERI*

4.1.1. Spatio-Temporal Characteristics of the TDI

The CAM method has been utilized to derive the TDI value in the YREB for the years 2007, 2013, and 2019. Arc GIS10.2 was applied to visualize the TDI in 2007, 2013, and 2019 and was classified into four levels to reveal the evolutionary characteristics of the TDI in the YREB (Figure 2).

Based on chronological evolution, the TDI value in the YREB experienced a significant increase, from 4.18 to 12.51, between 2007 and 2019, with a growth rate of 15.33% per annum. Across the regions, the average TDI of upstream, midstream, and downstream all exhibit a noticeable upward trend. Among them, the downstream experienced the most substantial growth, with a growth rate of 25.52% per year, while the growth rate of upstream and midstream was more moderate, at 16.86% and 12.61% per annum, respectively. Contrastingly, the downstream in the YREB exhibited a sturdy economic foundation, well-developed transport network, and prime market location. Consequently, the TDI in the downstream witnessed a more rapid growth rate compared to the midstream and upstream.

From Figure 2, it is found that: (1) the TDI of regional tourism hotspots such as Shanghai, Chongqing, Hangzhou, Nanjing, Chengdu, Wuhan, and Changsha were at a high level during the study period, while the TDI of regional fringe cities such as northern Jiangsu, southern Hubei, northeastern Yunnan, and southern Sichuan is generally low. This is mainly related to the city's tourism resource endowment, economic foundation, level of tourism facilities, and other factors; (2) The TDI of the YREB shows a more obvious spatial agglomeration feature, forming two tourism clusters—namely, the downstream tourism cluster, with Shanghai as the core, and the upstream tourism cluster, with Chongqing as the core. Meanwhile, the scope of the two tourism clusters is expanding, with the downstream tourism cluster expanding to Zhejiang and Jiangxi, and the upstream tourism cluster expanding to Sichuan and Guizhou.

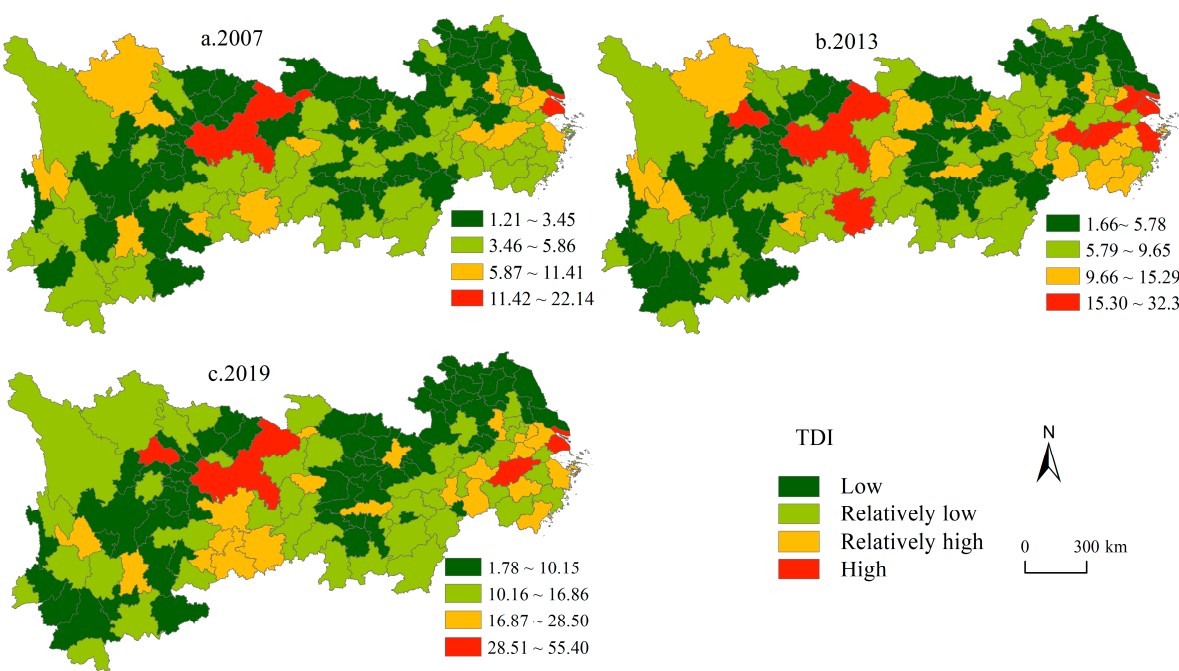

**Figure 2.** Spatial pattern of TDI in the YREB.

#### 4.1.2. Spatio-Temporal Characteristics of the ERI

The CAM method is applied to obtain the value of the ERI in the YREB in 2007, 2013, and 2019.

From the chronological evolution, the value of the ERI in the YREB declined during the study period, from 8.31 to 7.55, with a rate of decline of 1.45% per year, which implies an overall weakening trend in the ERI. The value of the ERI slowed down significantly, in stages, from 2013–2019, relative to the decline from 2007–2013, and its average annual decline decreased from 1.21% to 0.21%. In addition, the average ERI of the downstream began to show a significant upward trend. This indicates that the transformation of resource conservation in the YREB has slowed down the degradation of the urban eco-environment, especially in the downstream area, where the shift to green and high-quality urban development has achieved initial results.

With the help of ArcGIS10.2, the ERI is classified into four levels, and the results are shown in Figure 3. From the spatial pattern, the high-level cities in the YREB are mainly Shanghai in the downstream and Ganzi Prefecture in Sichuan in the upstream; the cities with a relatively high level, such as Suzhou, Wuxi, Changzhou, Nanjing, Jiaxing, Aba Prefecture, and Liangshan Prefecture, are distributed around the high-level cities, while the low and relatively low cities are primarily located in the midstream of the YREB. In general, the spatial pattern of "collapse in the middle" with "two high ends and low middle" is more obvious and has not changed substantially.

#### 4.2. Spatial Relationship between the TDI and ERI

The present paper employs the BISA method to uncover the global and local spatial correlation of the TDI and ERI in the YREB. The BI-GMSA results reveal that the global Moran's I of the TDI and ERI in the YREB in 2007, 2013, and 2019, stand at 0.13, 0.14, and 0.11, respectively. All of these findings are statistically significant and demonstrate the existence of significant positive spatial correlation between the TDI and ERI. The results strongly suggest that the TDI has enriched the ERI levels in the YREB. Meanwhile, the BI-LISA findings reveal the presence of four types of spatial clustering relationships between the TDI and ERI; namely, HH, HL, LH, and LL (Figure 4).

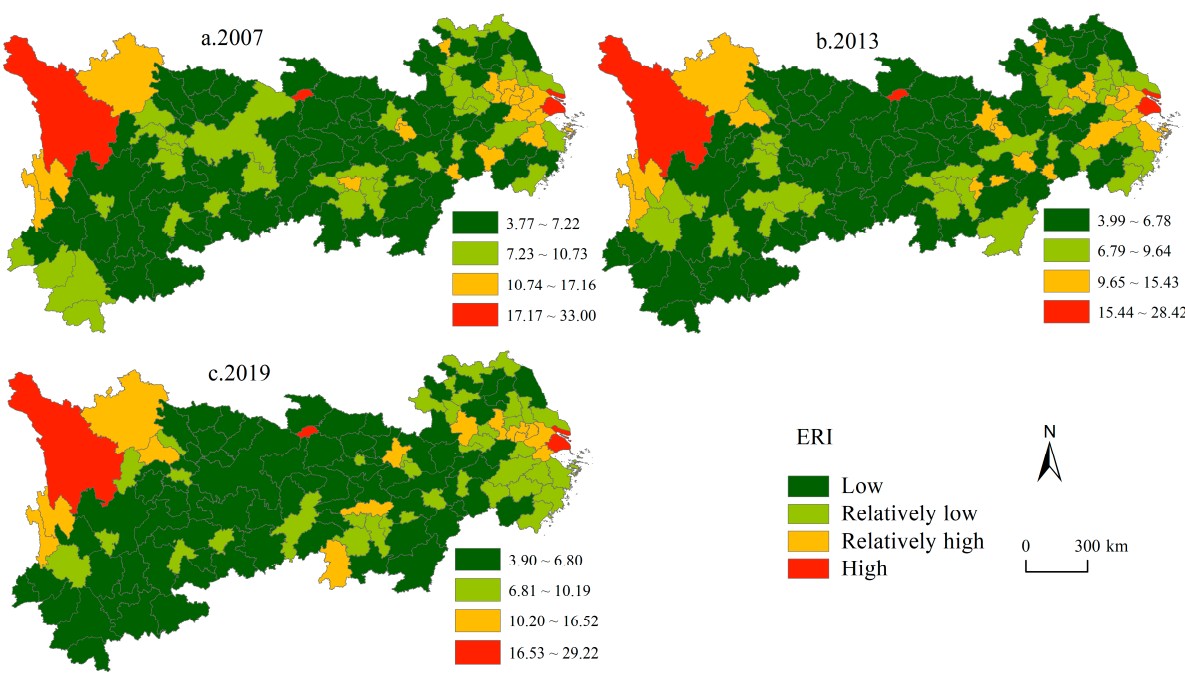

**Figure 3.** Spatial pattern of ERI in the YREB.

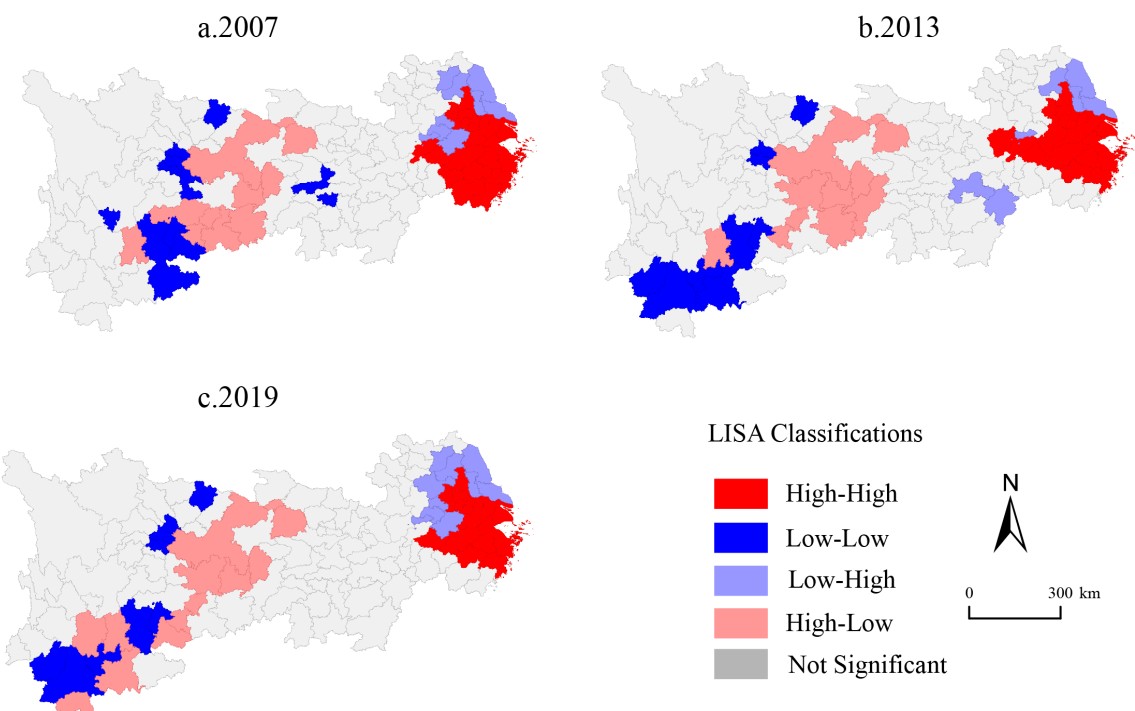

**Figure 4.** Bi-LISA results of TDI and ERI in the YREB.

The HH-type represents neighborhoods where both the TDI and ERI have high values and symbiotically promote each other's development. This type is primarily concentrated in Shanghai, Zhejiang, and southern Jiangsu regions. The growth of the region's urban tourism industry continues to shift from crude scale growth to intensive quality development, increasing the intensity of ecological resilience by reducing the disturbance to the ecological environment caused by tourism activities and product supply. The LH-type, on the other hand, is majorly distributed around HH-type areas, spanning central Jiangsu, northern Jiangsu, and southern Anhui. Hence, the TDI and ERI of the LH-type area display

a negative relationship in space. Compared with the HH-type areas, the TDI still has considerable room for improvement. Human activities such as the development of tourism projects and the increase in the scale of tourists brought about by tourism development put less pressure on the ecosystem, making the area ecologically resilient. The HL-type is predominantly found in Chongqing, eastern Guizhou, and western Hunan. This region has abundant tourism resources and rapid tourism economic growth, but also places tremendous pressure on its eco-environment, making the ecosystem less resilient over the study period. The LL-type, scattered around the HL-type areas, including Sichuan, Yunnan, etc., is primarily mountainous and boasts of sound eco-environmental conditions. Therefore, a major challenge is boosting both the ERI and TDI simultaneously. Overall, the HH-type and LH-type cities should focus on eco-environmental protection and promote the synergy of the TDI and ERI through a shift towards high-quality tourism development. In contrast, although the HL-type and LL-type cities have a good regional ecological base, the ERI is low, indicating that there is a need to improve ecological protection in the region, which should not jeopardize the ERI in favor of tourism economic growth, even though they have a more robust eco-environmental foundation.

It is important to note that the spatial correlation between the TDI and ERI is non-significant in some regions, indicating that the spatial relationship between the TDI and ERI in these regions has not yet formed a type of clustering.

### *4.3. The Effect of the TDI on the ERI*
#### 4.3.1. Model Construction

This paper constructs a model of the effect of the TDI on the ERI from the classical STIRPAT environmental impact model, which has the following standard form [77–79]:

$$I = \alpha P^b A^c T^d e \tag{13}$$

where *I* is the environmental variable, *P, A, T* are the demographic, economic, and technological variables, respectively; *b*, *c*, and *d* are the variable index values.

By adding the TDI as a variable in the STIRPAT model and taking logarithms on both sides to eliminate the effect of heteroskedasticity, the model of the effect of the TDI on the ERI is obtained, with the following equation:

$$lnERI = ln\alpha + \beta_1 lnTDI + \beta_2 lnPOP + \beta_3 lnGDP + \beta_4 lnOPEN + lne \tag{14}$$

where POP is the population density; GDP indicates the city affluence; OPEN is the external development level, expressed as the ratio of FDI to GDP.

#### 4.3.2. Overall Effect Analysis

This study employed OLS for model estimation (Table 2). The results show that Moran's I of the residuals is significantly positive in 2007, 2013, and 2019. In addition, its spatial-error term and spatial-lag term are also statistically significant. These findings suggest a noticeable spatial reliance of model residuals, leading to bias in the estimation results if spatial correlation attributes are disregarded. Remarkably, the $R^2$ of the SEM, SLM, and SEMLD all experience a significant improvement after accounting for spatial correlation. This is a clear indication that spatial econometric models with spatial factors incorporated outshine those without. Notably, when comparing the SEM, SLM, and SEMLD test results, the SEMLD generates relatively smaller values of both the Schwartz (*SC*) and Akaike information criterion (*AIC*) and a relatively larger likelihood (*LogL*). These results demonstrate the SEMLD model's superior capacity to model the TDI's impact on the ERI, making it the optimal model for this study.

**Table 2.** Spatial regression of the TDI on the ERI in the YREB.

| Variable | 2007 | | | | 2013 | | | | 2019 | | | |
|---|---|---|---|---|---|---|---|---|---|---|---|---|
| | OLS | SLM | SEM | SEMLD | OLS | SLM | SEM | SEMLD | OLS | SLM | SEM | SEMLD |
| lnTDI | 0.20 *** | 0.18 *** | 0.17 *** | 0.19 *** | 0.13 ** | 0.11 ** | 0.10 * | 0.12 ** | 0.20 *** | 0.19 *** | 0.24 *** | 0.20 *** |
| | (0.00) | (0.00) | (0.00) | (0.00) | (0.02) | (0.04) | (0.09) | (0.05) | (0.00) | (0.00) | (0.00) | (0.00) |
| lnPOP | 0.03 | 0.02 | 0.02 | 0.02 | 0.03 | 0.02 | 0.02 | 0.03 | 0.03 | 0.03 | 0.02 | 0.03 |
| | (0.30) | (0.40) | (0.48) | (0.43) | (0.28) | (0.31) | (0.32) | (0.33) | (0.19) | (0.23) | (0.30) | (0.25) |
| lnGDP | −0.01 | −0.03 | −0.04 | −0.03 | −0.05 | −0.05 * | −0.06 * | −0.05 * | −0.01 | −0.02 | −0.04 | −0.02 * |
| | (0.72) | (0.36) | (0.31) | (0.45) | (0.16) | (0.08) | (0.08) | (0.08) | (0.69) | (0.45) | (0.22) | (0.08) |
| lnOPEN | 0.05 ** | 0.04 * | 0.03 | 0.05 * | 0.10 *** | 0.09 *** | 0.10 *** | 0.10 *** | 0.04 * | 0.03 | 0.02 | 0.08 ** |
| | (0.04) | (0.09) | (0.21) | (0.06) | (0.00) | (0.00) | (0.00) | (0.00) | (0.06) | (0.14) | (0.30) | (0.03) |
| Spatial-lag | | 0.36 *** | | 0.35 *** | | 0.34 *** | | 0.35 *** | | 0.36 *** | | 0.37 *** |
| | | (0.00) | | (0.00) | | (0.00) | | (0.00) | | (0.00) | | (0.00) |
| Spatial-err | | | 0.36 *** | 0.36 *** | | | 0.34 *** | 0.34 *** | | | 0.43 *** | 0.41 *** |
| | | | (0.00) | (0.00) | | | (0.00) | (0.00) | | | (0.00) | (0.00) |
| Constant | 1.91 *** | 1.31 *** | 2.09 *** | 1.87 *** | 2.34 *** | 1.73 *** | 2.50 *** | 2.33 *** | 1.56 *** | 0.96 *** | 1.68 *** | 1.77 *** |
| | (0.00) | (0.00) | (0.00) | (0.00) | (0.00) | (0.00) | (0.00) | (0.00) | (0.00) | (0.01) | (0.00) | (0.00) |
| Moran's I | 2.89 *** | | | | 3.37 *** | | | | 4.14 *** | | | |
| | (0.00) | | | | (0.00) | | | | (0.00) | | | |
| LM (lag) | 11.96 *** | | | | 9.63 *** | | | | 14.96 *** | | | |
| | (0.00) | | | | (0.00) | | | | (0.00) | | | |
| Robust LM (lag) | 10.26 *** | | | | 0.98 | | | | 1.52 | | | |
| | (0.00) | | | | (0.32) | | | | (0.22) | | | |
| LM (error) | 6.33 *** | | | | 8.66 *** | | | | 13.47 *** | | | |
| | (0.01) | | | | (0.00) | | | | (0.00) | | | |
| Robust LM (error) | 4.62 ** | | | | 0.02 | | | | 0.02 | | | |
| | (0.03) | | | | (0.89) | | | | (0.87) | | | |
| LM (lag and error) | 16.58 *** | | | | 9.65 *** | | | | 14.99 *** | | | |
| | (0.00) | | | | (0.00) | | | | (0.00) | | | |
| R2 | 0.13 | 0.23 | 0.22 | 0.22 | 0.15 | 0.22 | 0.22 | 0.23 | 0.16 | 0.26 | 0.27 | 0.27 |
| LogL | −55.45 | −50.02 | −50.64 | −52.32 | −38.04 | −33.48 | −33.89 | −33.56 | −38.97 | −32.87 | −32.41 | −32.33 |
| AIC | 120.90 | 112.03 | 115.29 | 118.94 | 86.08 | 78.95 | 77.78 | 76.57 | 87.94 | 77.75 | 74.81 | 73.81 |
| SC | 135.24 | 129.24 | 135.36 | 133.26 | 100.42 | 96.16 | 92.12 | 89.43 | 102.28 | 94.95 | 89.15 | 88.67 |
| Obs. | 130 | 130 | 130 | 130 | 130 | 130 | 130 | 130 | 130 | 130 | 130 | 130 |

Note: *, ** and *** indicate significance at the level of 10%, 5% and 1%, respectively.

Based on the findings of the SEMLD, *LnTDI*'s regression coefficients are significantly positive, indicating that the TDI generally bolsters the ERI. This can be attributed to two key factors. Firstly, the TDI can incentivize destination cities to focus on maintaining and enhancing eco-environmental standards while ensuring greater ecosystem recovery and capacity. Secondly, the TDI can assist in optimizing the industrial structure of destinations and augmenting the adaptability of urban ecosystems. The regression coefficient of *LnTDI*, upon its initial decrease, indicates that an exclusive focus on the tourism industry's economic functions has an insignificant impact on the ERI performance. Hence, it is imperative to prioritize the value of the multifaceted ecological and social functions of the TDI.

Furthermore, it is worth noting that the spatial-lag is significantly positive, thereby confirming the presence of a spatial spillover effect on the ERI. The urban ERI is expected to expand by approximately 0.35% for every 1% increase in the ERI of neighboring cities. This, in turn, results in an ERI growth "gift" from the neighbors [80]. Several factors contribute to this phenomenon, including the ERI's reflection of the intricate combination of ecosystem resistance, adaptability, and recovery. Moreover, a positive ERI development has a constructive impact on neighboring cities' ecosystems through the emulation and diffusion of technological innovations. Correspondingly, the spatial-err of the SEMLD models is also significantly positive, proving that the ERI in the YREB region is not only influenced by the TDI, but also other factors, such as the population density and economic growth.

### 4.3.3. Heterogeneity Analysis of the Effect

SEMLD, a spatial econometric model, provides a global perspective for analyzing the impact of the TDI on the ERI; however, it fails to capture spatial differences in the impacts. To reveal hidden local differences within the overall regression results, this paper employs GWR to estimate the spatio-temporal heterogeneity of the TDI's impact on the ERI. According to the test results, the $R^2$ values range between 0.471 and 0.620, marking a significant improvement over the OLS $R^2$ values (0.118–0.165). Meanwhile, metrics such as *AICc, AIC,* and *SSE* demonstrate significant reductions when compared with OLS, suggesting that GWR has a superior explanatory power for spatio-temporal heterogeneity estimation.

This paper utilizes ArcGIS 10.2 and the Jenks natural breakpoint method to visualize the GWR estimation outcomes. This technique provides detailed information on the TDI's effect intensity on the ERI, as well as its spatio-temporal variation in 2007, 2013, and 2019 (Figure 5). The estimated coefficients' sign and magnitude indicate the direction and intensity of the TDI's impact on the ERI, respectively.

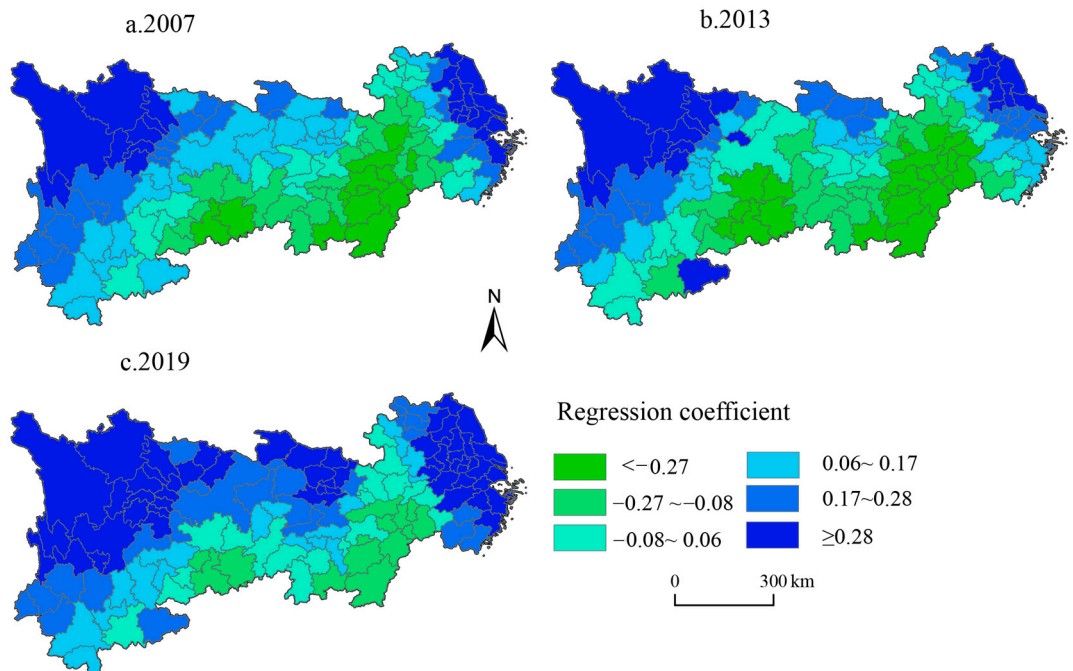

**Figure 5.** The spatial pattern of regression coefficient of the TDI on the ERI.

The estimation results indicate that the TDI has evident spatio-temporal heterogeneity on the ERI in the YREB, exhibiting both positive facilitative and negative inhibitory effects. From the spatial pattern, the regions with positive effects are concentrated in Yunnan and Sichuan in the upstream and the Yangtze River Delta (YRD) in the downstream. The range of regions with high-intensity positive effects is expanding continuously over time. Regions with negative inhibitory effects are concentrated in Jiangxi, Hubei, and Hunan in the midstream, and this region exhibits a more distinct narrowing trend. This suggests that cities in the YREB not only focus on the scale growth of tourism development, but also emphasize tourism's comprehensive benefits. Consequently, they strive to promote the capacity of urban ecosystems to withstand pressure from quality development of tourism and to adjust their ability to recover.

## 5. Discussion

This study sought to measure the combined impact of the TDI on the ERI in an integrated manner. Quantitative research methods such as spatial econometric modeling and geographically weighted regressions were used to address two research questions.

The first question is what is the development of the TDI and ERI in the YREB? From the results of the assessment, the TDI in the YREB increased significantly between 2007 and 2019, with an annual growth rate of 15.33%. This increase can be attributed to factors such as tourism resource endowment and socioeconomic development, which are supported by previous studies [81–85]. Moreover, as Min (2015) and Xiao et al. (2022) point out, socio-economic factors such as transportation, economy, and urbanization are the main drivers behind TDI growth [32,65,86,87].

The YREB is an important industrial agglomeration in China and has contributed significantly to the Chinese economy. However, this economic growth has come at a cost, as the environmental quality has suffered [87,88]. The results confirm this claim, with the ERI of the YREB declining overall by 1.45% per year. Nevertheless, the downward trend of the ERI decelerated significantly between 2013 and 2019, especially in the downstream, where the ERI has improved considerably. This achievement is attributed to national policies that prioritize ecological protection, especially along the YRD. Since 2013, China's State Council has issued several policy documents related to environmental protection along the YREB. The implementation of these policies has slowed the unabated deterioration of the ERI and stimulated the transformation of the YRD to a resource-efficient and eco-friendly development approach [84,85,87].

With regard to the second question (Is the impact of the TDI on the ERI positive or negative?), this study empirically tested the spatio-temporal heterogeneity of this impact at the meso level. In previous studies [30,33], there has been disagreement on whether the impact of tourism development on the ERI is positive or negative. This study resolves this disagreement and finds that, within one region, the impact of the TDI on the ERI can be both positive and inhibitory, which mainly stems from the differences in the direction of tourism development in tourist destinations. Tourism destinations that prioritize ecological impacts, supported by the concept of sustainable development, see a positive impact of the TDI on the ERI [87,89]. On the contrary, the TDI has a dampening effect on the ERI in certain tourist places that only emphasize the expansion of the tourism economy while neglecting ecological and environmental protection [84,87].

Meanwhile, it has been observed that, over time, the areas where the TDI generates catalytic effects on the ERI are expanding, whereas the regions where it has an inhibitory effect are diminishing. This spatio-temporal heterogeneity implies that an increasing number of cities in the YREB are now prioritizing eco-environmental protection over merely industrial scale and economic growth while developing tourism [87]. This has inevitably fostered a co-ordinated development between the TDI and ERI [85], on one hand by enhancing the eco-environment's aesthetics and optimizing it to create a welcoming tourist destination [38,88], and on the other hand, by utilizing tourism to enhance the ecosystem's resilience and adaptability to withstand internal and external pressures [32,39,44].

In addition, consistent with the existing research literature [6,13,27], this study also found a significant positive spatial spillover effect of the ERI. This spatial spillover of the ERI suggests that the ERI is positively influenced by neighboring regions. The positive spatial spillover effect of the ERI may be attributed to the environmental protection policy of the YREB [87], technological innovations in environmental protection and its knowledge dissemination [80], and the demonstration effect of environmental protection performance [25,42].

## 6. Conclusions

### 6.1. Main Conclusions

This paper uses BISA, SEMLD, and GWR to empirically reveal the spatio-temporal heterogeneity of the TDI's impact on the ERI in the YREB. It presents the following conclusions:

Firstly, during the study period, the growth of the TDI in the YREB was more pronounced, particularly in the YRD. Concerning the spatial distribution, Shanghai, Chongqing, Hangzhou, Nanjing, Chengdu, Wuhan, Changsha, and other regional center cities have a high-level TDI, forming a tourism industry cluster in the downstream with Shanghai as

the core, and in the upstream, with Chongqing as the core, and the scope of the clusters is expanding to the surrounding areas.

Secondly, while the ERI showed a declining trend in the YREB, the decline slowed significantly. This trend generated a "central collapse" pattern of "high at both sides and low in the center", with cities having high ERI levels primarily clustered downstream and in the upstream western Sichuan region.

Thirdly, the bivariate spatial autocorrelation results show that there is a clear spatial dependence between the TDI and ERI in the YREB. In terms of spatial types, the HH-type and LH-type are mainly distributed in the YRD, which pay more attention to the synergistic development of the TDI and ERI; the HL-type and LL-type are mainly concentrated in the upstream, which has a good ecological base and strong ecological vulnerability, and the rapid development of tourism exerts greater pressure on the ecological system.

Finally, overall, the urban TDI will promote its ERI, but there is significant spatial heterogeneity in its impact. The regions with positive impacts are mainly concentrated in Yunnan and Sichuan in the upstream and the YRD in the downstream, while the regions with negative inhibitory effects are mainly concentrated in Jiangxi, Hubei, and Hunan in the midstream, and the positive impacts are continuously strengthened and the negative impacts are weakened through the comprehensive benefits of the tourism industry.

### 6.2. Theoretical Contributions

The study conducted in this paper enriches the current literature on the relationship between tourism and ecosystems.

This study not only confirms the existence of a dichotomous contradiction in the impact of tourism on ecosystems, but also finds that this contradiction is mainly due to differences in the direction of urban tourism development. If cities prioritize sustainability in developing tourism, the TDI has a significant positive impact on the ERI. On the contrary, if only the growth of the tourism economy is emphasized and the protection of ecological environment is neglected, the TDI will have a significant inhibitory effect on the ERI.

### 6.3. Policy Implications

To commence, cities in the YREB must prioritize the high-quality promotion of the TDI. This can be achieved by embracing the concept of high-quality development, promoting the transformation of the TDI, implementing resource conservation and carbon emission reduction methods, building a modern urban tourism sector system, exploiting the eco-environmental effects of the TDI, and continuously improving the ERI.

Secondly, classified policies should be established to promote the positive effect of the TDI on the ERI. The region should focus on strengthening the promotion of the TDI to the ERI in the YRD and Southwest China, continuing the optimization of the industrial structure, and improving the green development level of urban agglomerations in the midstream. Moreover, the eco-environmental efficiency of urban tourism development should be enhanced to propel the transformation of the eco-environmental effect of the TDI from inhibition to promotion.

Lastly, it is imperative to jointly develop and upgrade the ERI of the YREB. This can be achieved by strengthening cross-city joint environmental governance, enhancing regions with a low ERI to cope with risks, and improving the adaptive and recovery capacity of ecosystems. The goal is to jointly build an ecological security pattern along the YREB in a sustainable manner.

### 6.4. Limitations and Avenues for Future Research

There are still certain limitations of this paper that require clarification. Firstly, concerning the selection of indicators for the TDI and ERI, the absence of statistical indicators in some cities has restricted their selection, potentially influencing the results. Secondly, this paper has only analyzed data from 2007, 2013, and 2019 due to the excessive amount

of data, which may not accurately reflect the evolutionary trajectory and impact trajectory of the TDI and ERI.

Future research can focus on a certain tourist place to explore the changing law of the TDI's impact on the ERI in the process of tourism development and reveal its impact mechanism. It can also select case places with different tourism development modes to explore the differences in the impact of the TDI on the ERI through comparative studies. In addition, future research can also focus on the mediating effect of the impact of the TDI on the ERI, revealing its indirect impact through the selection of mediating variables.

**Author Contributions:** Conceptualization, K.W. and X.Z.; Data curation, X.C. and Z.L.; Funding acquisition, K.W.; Investigation, S.Z.; Software, Z.L.; Visualization, X.C.; Writing—original draft, K.W.; Writing—review and editing, X.Z. All authors have read and agreed to the published version of the manuscript.

**Funding:** This research was funded by the National Natural Science Foundation of China (No. 41961025).

**Institutional Review Board Statement:** Not applicable.

**Informed Consent Statement:** Not applicable.

**Data Availability Statement:** The datasets used in this research are available upon request.

**Conflicts of Interest:** The authors declare no conflict of interest.

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
