# Peer review of "The Effects of Tourism Development on Eco-Environment Resilience and Its Spatio-Temporal Heterogeneity in the Yangtze River Economic Belt, China"

_sustainability, doi:10.3390/su152216124_

Round 1
Reviewer 1 Report (Previous Reviewer 1)
Comments and Suggestions for Authors
I accept paper in present form.
Author Response
Thank you very much for reviewing the article!
Reviewer 2 Report (Previous Reviewer 2)
Comments and Suggestions for Authors
The changes I suggested in my comments have been made.
Comments on the Quality of English LanguageMinor editing of English language required
Author Response
Thank you very much for reviewing the article!We checked the language issue and we'll continue to refine it later.
Reviewer 3 Report (Previous Reviewer 3)
Comments and Suggestions for Authors
I maintain the same remarks made to the previous version. I believe that the approach used is not very useful for the problem raised.
Author Response
Thank you so much for your review!Please see the attachment.

Reviewer 4 Report (Previous Reviewer 4)
Comments and Suggestions for Authors
Paper can be accepted. Before that, I also need to state that the authors are advised to revise Figure 2. to use Chinese maps with review numbers as required by the Ministry of Natural Resources of China. This is a non-mandatory modification suggestion.
Comments on the Quality of English LanguageFormat and language need improvement.
Author Response
Thank you so much for your review! Please see the attachment.

Reviewer 5 Report (New Reviewer)
Comments and Suggestions for Authors This paper analyzes the effect mechanism of tourism development on eco-environment resilience in in the Yangtze River Economic Belt (YREB) based on the tourism development index (TDI) and eco-environment resilience index (ERI).The paper is well structured and well-illustrated. It has a sound methodology which is described in section 2. The introduction includes a figure which I think may fit better in section 2.
There are only some improvements which need to be done.
The paper has several paragraphs colored in red and blue. This issue needs to be addressed.
The discussion may be improved in terms of comparing the results with previous studies on the same theme.
The authors provided policy implications but I think that some theoretical implications need to be outlined as well.
Author Response
Thank you so much for your review! Please see the attachment.

This manuscript is a resubmission of an earlier submission. The following is a list of the peer review reports and author responses from that submission.
Round 1
Reviewer 1 Report
Comments and Suggestions for Authors
Thank you for reviewing possibility of paper titled The effects of tourism development on eco-environment resilience and its spatiotemporal heterogeneity in the Yangtze River Economic Belt, China. It's interesting stuty, however nedds some improvements:
- I suggest to add some references about urban/cities resilience in lines 44-55. A different model of urban resiliences authors can also find here DOI: 10.2478/mgrsd-2019-0028. urban areas also have resilience for crisises what is important issue in modern fluctuating economy, especially in current covid pandemia. I suggest to include some concepts about that: doi: 10.3986/AGS.7416, doi: 10.1007/s11769-017-0850-5
-In the Introduction (lines 84-86) I suggest to add some references on the impact of tourism on society (e.g. increase of jobs, improve of intrastructure, etc.) eg. https://doi.org/10.3390/land10121342
- The goal of the paper is clear, but authors should state what new they add to existing knowledge.
- line 241: Table 3 or 2?
- Figures 3,4,5: I understand that figure shows values changes during the years. It would be better to stay with the same border values for TDI (veery low, low...) for all years. It will visualize changes on figure.
- I see that authors love shortcuts, but sometimes they make the paper unreadable. It's very hard to read eg. chapter 3.2 or Conclusions. Conclusions should be understandable for all readers without studying all manuscript for shortcuts meaning.
Good luck wit the paper!
Reviewer 2 Report
Comments and Suggestions for Authors
REVIEWER’S EVALUATION SHEET
Manuscript number: sustainability-2425113
Manuscript’s title: The effects of tourism development on eco-environment resilience and its spatiotemporal heterogeneity in the Yangtze River Economic Belt, China
Confidential Comments to the Editor
The manuscript explores an intriguing research direction by investigating the effect mechanism of tourism development (TDI) on eco-environment resilience (ERI) in the Yangtze River Economic Belt (YREB). The authors have demonstrated professionalism and academic rigor in implementing this idea, indicating that they have put significant effort into their work. I would like to extend my congratulations to them for their commendable efforts. In my opinion, the manuscript meets the high standards required for publication in Sustainability, and I have only a few minor comments to offer.
Recap
The manuscript employs several analytical tools, including the Evaluation Indicator System for TDI and ERI, a Comprehensive Assessment Model, Bivariate Spatial Autocorrelation Analysis, Spatial Econometric Model, and Geographically Weighted Regression. These methodologies are utilized to investigate the effect mechanism of tourism development (TDI) on eco-environment resilience (ERI) in the Yangtze River Economic Belt (YREB). By conducting the analysis for the years 2007, 2009, and 2019, the study aims to offer valuable insights into this subject matter.
Detailed Comments
Introduction
[1] The abbreviation "I" in TDI and ERI was not immediately clear to me until reaching line 146, where it is explained as Infrastructure. To avoid confusion, it would be helpful to introduce and define the abbreviations TDI (Tourism Development Index) and ERI (Eco-Environment Resilience Index) earlier in the manuscript.
[2] In the sentence "It focused mainly on the assessment and influencing factors of the ERI," (line 57) it would be beneficial to provide a reference.
[3] The transition from paragraphs in lines 43-67, which focus on ERI, to suddenly introducing TDI seems somewhat abrupt. It would be beneficial to provide a more gradual transition and introduce TDI earlier in the section, offering some background or context on the topic.
[4] In line 76, the statement "Thirdly, the most discussed issue among scholars is the impact brought by tourists" raises the question of the source or reference that supports the claim that it is the most discussed issue.
[5] While Figure 1 appears to present significant and noteworthy findings, it does not receive sufficient emphasis in the manuscript. It would be beneficial to elaborate on the key insights or notable aspects highlighted in Figure 1. This will help to emphasize the importance of the figure and its contribution to the overall study.
Materials and Methods
[6] In the second paragraph of section 2.1 (lines 120-127), it would be beneficial to include specific numbers or statistics to provide empirical evidence that supports the description provided. This will enhance credibility.
[7] In line 140, the statement "Therefore, the years of 2007, 2013, and 2019 are selected as the timepoints in this paper" lacks an explanation for the specific selection of these dates. It would be helpful to provide a rationale or justification for choosing these particular years, such as their relevance to the research topic or the availability of data.
[8] In line 144, the phrase "Tourism, as a prototypical modern service industry" and in series 471 "building a modern urban tourism industry system" refers to the term "tourism industry." While the term "tourism industry" is widely used, it can be argued that "tourism sector" is a more accurate and appropriate terminology. Consider replacing "tourism industry" with "tourism sector" or using alternative terms that better reflect the nature of tourism.
[9] It would be beneficial to provide proper citations or references for each variable mentioned in Table 1. Including the sources of the variables will ensure transparency and allow readers to access additional information if needed.
Results
[10] The title of Table 2 stating "The average TDI of the YREB from 2007 to 2019" is incorrect since it presents the values for the specific years 2007, 2013, and 2019. The title should be revised to accurately reflect the contents of the table. The same applies to Table 3.
[11] In line 276, it is stated that "From the chronological evolution, the value of the ERI in the YREB declined during the study period, from 9.31 to 7.55." However, according to Table 3, the value is reported as 8.31 and not 9.31.
[12] It is unclear from the manuscript which areas are referred to as upstream, midstream, and downstream. To enhance clarity, it would be helpful to provide a clear definition or description of these terms and specify the geographical regions to which they correspond. This will ensure that readers can easily understand the spatial context being discussed.
Discussion
[13] In line 486, it is mentioned, "Firstly, concerning the selection of indicators for the TDI and ERI, the absence of statistical indicators in some cities has restricted the selection of indicators, potentially influencing the results." This point requires further analysis. It would be beneficial to provide additional explanations regarding which cities lack data and why this data is missing. Understanding the specific cities affected and the reasons behind the missing data will contribute to a more comprehensive discussion.
[14] In line 488, it states, "Secondly, this paper has only analyzed data from 2007, 2013, and 2019 due to the abundance of data, which may not accurately reflect the evolutionary trajectory and impact trajectory of the TDI and ERI." The phrase "due to the abundance of data" is unclear and seems contradictory to the statement. It would be helpful to clarify the intention of this statement and provide further explanation.
I hope that my comments will be helpful to the authors.
Sincerely,
Comments on the Quality of English LanguageREVIEWER’S EVALUATION SHEET
Manuscript number: sustainability-2425113
Manuscript’s title: The effects of tourism development on eco-environment resilience and its spatiotemporal heterogeneity in the Yangtze River Economic Belt, China
Confidential Comments to the Editor
The manuscript explores an intriguing research direction by investigating the effect mechanism of tourism development (TDI) on eco-environment resilience (ERI) in the Yangtze River Economic Belt (YREB). The authors have demonstrated professionalism and academic rigor in implementing this idea, indicating that they have put significant effort into their work. I would like to extend my congratulations to them for their commendable efforts. In my opinion, the manuscript meets the high standards required for publication in Sustainability, and I have only a few minor comments to offer.
Recap
The manuscript employs several analytical tools, including the Evaluation Indicator System for TDI and ERI, a Comprehensive Assessment Model, Bivariate Spatial Autocorrelation Analysis, Spatial Econometric Model, and Geographically Weighted Regression. These methodologies are utilized to investigate the effect mechanism of tourism development (TDI) on eco-environment resilience (ERI) in the Yangtze River Economic Belt (YREB). By conducting the analysis for the years 2007, 2009, and 2019, the study aims to offer valuable insights into this subject matter.
Detailed Comments
Introduction
[1] The abbreviation "I" in TDI and ERI was not immediately clear to me until reaching line 146, where it is explained as Infrastructure. To avoid confusion, it would be helpful to introduce and define the abbreviations TDI (Tourism Development Index) and ERI (Eco-Environment Resilience Index) earlier in the manuscript.
[2] In the sentence "It focused mainly on the assessment and influencing factors of the ERI," (line 57) it would be beneficial to provide a reference.
[3] The transition from paragraphs in lines 43-67, which focus on ERI, to suddenly introducing TDI seems somewhat abrupt. It would be beneficial to provide a more gradual transition and introduce TDI earlier in the section, offering some background or context on the topic.
[4] In line 76, the statement "Thirdly, the most discussed issue among scholars is the impact brought by tourists" raises the question of the source or reference that supports the claim that it is the most discussed issue.
[5] While Figure 1 appears to present significant and noteworthy findings, it does not receive sufficient emphasis in the manuscript. It would be beneficial to elaborate on the key insights or notable aspects highlighted in Figure 1. This will help to emphasize the importance of the figure and its contribution to the overall study.
Materials and Methods
[6] In the second paragraph of section 2.1 (lines 120-127), it would be beneficial to include specific numbers or statistics to provide empirical evidence that supports the description provided. This will enhance credibility.
[7] In line 140, the statement "Therefore, the years of 2007, 2013, and 2019 are selected as the timepoints in this paper" lacks an explanation for the specific selection of these dates. It would be helpful to provide a rationale or justification for choosing these particular years, such as their relevance to the research topic or the availability of data.
[8] In line 144, the phrase "Tourism, as a prototypical modern service industry" and in series 471 "building a modern urban tourism industry system" refers to the term "tourism industry." While the term "tourism industry" is widely used, it can be argued that "tourism sector" is a more accurate and appropriate terminology. Consider replacing "tourism industry" with "tourism sector" or using alternative terms that better reflect the nature of tourism.
[9] It would be beneficial to provide proper citations or references for each variable mentioned in Table 1. Including the sources of the variables will ensure transparency and allow readers to access additional information if needed.
Results
[10] The title of Table 2 stating "The average TDI of the YREB from 2007 to 2019" is incorrect since it presents the values for the specific years 2007, 2013, and 2019. The title should be revised to accurately reflect the contents of the table. The same applies to Table 3.
[11] In line 276, it is stated that "From the chronological evolution, the value of the ERI in the YREB declined during the study period, from 9.31 to 7.55." However, according to Table 3, the value is reported as 8.31 and not 9.31.
[12] It is unclear from the manuscript which areas are referred to as upstream, midstream, and downstream. To enhance clarity, it would be helpful to provide a clear definition or description of these terms and specify the geographical regions to which they correspond. This will ensure that readers can easily understand the spatial context being discussed.
Discussion
[13] In line 486, it is mentioned, "Firstly, concerning the selection of indicators for the TDI and ERI, the absence of statistical indicators in some cities has restricted the selection of indicators, potentially influencing the results." This point requires further analysis. It would be beneficial to provide additional explanations regarding which cities lack data and why this data is missing. Understanding the specific cities affected and the reasons behind the missing data will contribute to a more comprehensive discussion.
[14] In line 488, it states, "Secondly, this paper has only analyzed data from 2007, 2013, and 2019 due to the abundance of data, which may not accurately reflect the evolutionary trajectory and impact trajectory of the TDI and ERI." The phrase "due to the abundance of data" is unclear and seems contradictory to the statement. It would be helpful to clarify the intention of this statement and provide further explanation.
I hope that my comments will be helpful to the authors.
Sincerely,
Reviewer 3 Report
Comments and Suggestions for Authors
The impacts of tourism activity on the territory are of undoubted interest.
However, the empirical approach and the territorial dimension chosen is so broad that the conclusions are not in line with the objectives.
Suggestions:
1. The area of analysis is very broad. The index tourism pressure works on specific destinations and in smaller areas. The macro-analysis done in the paper can be used to choose a specific area and should be studied in greater detail.
2. In relation to the above, it is not clear from the variables used whether tourism development is the cause of the pressure or whether urbanization processes are.
3. The TDI variables approximate the tourism phenomenon, but do not show a direct relationship between the tourism phenomenon and the carrying capacity of the destination. For example, the number of tourists received over the total population, hotel beds over the total population, etc.
4. Other variables that approximate the tourism phenomenon are very imperfect, such as employment in services. They are not well approximated to the tourism phenomenon other than urbanisation itself.
5. The spatial correlation analysis carried out would allow, as a previous step, to choose those areas or destinations where there is a clear relationship between TDI and ERI, which is the objective of the work. The analysis of spatial correlation is well done but the results are not linked with the main papers objectives.
6. Subsequently, an in-depth analysis of the chosen areas should be carried out with a set of indicators that are more appropriate for analysing the pressure of tourism on the destination.
7. The spatial correlation analysis, as a previous work, can be used to choose two areas type. One with h-h correlation TDI ERI and proof of a casual relationship with a set of variables that can be measured the pressure of possible over-tourism effect, in contrast with other destinations with l-l correlation.
8. The conclusions are very poor and reach common ground. They are NOT in-depth
Reviewer 4 Report
Comments and Suggestions for Authors
1. lines 282-283: The Results and analysis section should be more concise and analyze only your findings
2. Part 2.1, 2.2: Authors need to cite information about the field of study, the procedures used, and the statistical analysis used to make it clearer.
3. Part 2.3.1: The language should be more concise and only talk about your evaluation system.
4. line 463-456: There was a significant correlation in some areas, and a negative correlation or no significant correlation in others. What is the reason for the difference in the car building part?
5. lines 469-470: the author should remove the general statements and the discussioon should related to their specific findings.
6. page 2, line 548-549: “whose tourism development should avoid sacrificing or reducing ecological resilience” The results section I think we should just talk about what are the results of your research.
Round 2
Reviewer 1 Report
Comments and Suggestions for Authors
Authors have made huge effort to improve the paper. I accept the paper in present form.
Author Response
Thank you to the reviewer for their review.
Reviewer 3 Report
Comments and Suggestions for Authors
The suggestions made in the first evaluation have been answered by the authors. All of them will be taken into account in future research. It does not change, therefore, the main criticisms made in the first version.
The analysis carried out does not allow an adequate assessment of the phenomenon of tourist pressure on destinations. Its conclusions are very aggregated (macro level) and are of little use for the initial objectives.
Round 3
Reviewer 3 Report
Comments and Suggestions for Authors
The authors' responses to the comments made in the previous version do not allow for a change of opinion.
The comments made have not been taken into account either because of a lack of statistical information or because they require a different methodological approach.
The authors recognize the value of these comments but decide to undertake the changes in future research, which implies that the limitations considered by this reviewer are still present in the paper.